# Landmark Distance Impacts the Overshadowing Effect in Spatial Learning Using a Virtual Water Maze Task with Healthy Adults

**DOI:** 10.3390/brainsci13091287

**Published:** 2023-09-05

**Authors:** Róisín Deery, Seán Commins

**Affiliations:** Psychology Department, Maynooth University, W23 F2K8 Kildare, Ireland

**Keywords:** cue competition, overshadowing, spatial learning, memory, virtual water maze

## Abstract

Cue competition is a key element of many associative theories of learning. Overshadowing, an important aspect of cue competition, is a phenomenon in which learning about a cue is reduced when it is accompanied by a second cue. Overshadowing has been observed across many domains, but there has been limited investigation of overshadowing in human spatial learning. This experiment explored overshadowing using two landmarks/cues (at different distances to the goal) in a virtual water maze task with young, healthy adult participants. Experiment 1 initially examined whether the cues used were equally salient. Results indicated that both gained equal control over performance. In experiment 2, overshadowing was examined using the two cues from experiment 1. Results indicated that overshadowing occurred during spatial learning and that the near cue controlled searching significantly more than the far cue. Furthermore, the far cue appeared to have been completely ignored, suggesting that learning strategies requiring the least amount of effort were employed by participants. Evidence supporting an associative account of human spatial navigation and the influence of proximal cues was discussed.

## 1. Introduction

Navigating to a particular place and subsequently recalling its location is a fundamental life skill. Remembering where food is hidden or how to return to a breeding site is often a matter of life and death for migrating animals. For humans too, recalling places and finding our way is of critical importance. It is only when people become disorientated, as in the case with Alzheimer’s disease [1], or are lost in low-visibility conditions, that we understand the extent of our reliance on spatial memory for everyday activities. One influential account of spatial navigation and memory is the cognitive map theory [2,3]. In this model, animals can develop a map-like representation through their exploration of an environment via the encoding of relationships between landmarks, between landmarks and the goal, and between the navigator, landmarks, and the goal. From this, novel routes and shortcuts to the goal can be created. While the cognitive map has received widespread attention across a range of species (birds, [4] insects [5] (but see [6] and mammals [7]), an alternative account of spatial learning based on associative learning theory (e.g., see [8]) may provide a better and arguably simpler explanation. With this model, only certain elements of the environment are learned and associated, and the association is formed only when required; as such, there is no need to develop a full map of the environment. All landmarks/cues are therefore not treated equally, with some given more weight or relevance compared to others.

The salience, or “significance or noticeability”, of a cue [9] (p. 340), can provide a particular cue with relevance. Specific features of a cue, such as its size, shape, and luminescence, can all contribute to its salience. For example, [10] demonstrated that rats trained with brighter cues outperformed those with dimmer cues regardless of how near the cue was to the goal in the Morris water maze. However, proximity to the goal is a very important salient feature, with many studies showing that animals learn and recall locations better when provided with proximal cues compared to distal ones (honeybees: [11]; birds: [12]; rats: [13]; mice: [14]; humans: [15,16]). Such competition between cues (near vs. far; bright vs. dim; large vs. small) is a hallmark of associative learning theory and can be tested more formally using blocking and overshadowing designs [17,18]. Blocking refers to the observation that pre-training with one cue of a compound subsequently interferes with learning the second cue of the compound. Whereas overshadowing refers to the idea that individual cues in a compound will share the associative value, and therefore learning will be reduced for each component cue compared to if one of those cues was learned individually [19]. Overshadowing has been previously observed with human participants across many domains, including visual and tactile recognition [20], verbal learning [21], geometrical learning [22], and causal learning [23].

Overshadowing has also been observed in the spatial domain across many studies with nonhuman animals [10,14,24,25,26,27]. However, research with human participants has not been examined to the same extent [28], and this research has tended to focus on the debate on whether cue competition (including overshadowing) can be applied to all spatial elements, particularly with respect to boundaries and the geometry [11,29] of the testing environment (for example, see [30,31,32]). Other factors, including the role of landmark salience (size, shape, proximity, etc.) on overshadowing in human participants, have received less attention. Indeed, a study by Herrera et al. [33] recently emphasised the importance of proximity by demonstrating the overshadowing of geometry by near but not distal landmarks. Importantly, the authors suggested that contiguity in space (i.e., closeness) is the key factor, and it is this feature that allows for overshadowing to take place. However, in a study examining competition among landmarks in a virtual water maze task, Sansa et al. [28] reported an overshadowing effect with both distal and proximal landmarks. This observation is not consistent with Herrera et al.’s suggestion, where you might expect to see overshadowing with proximal cues only and not distal ones. In both of these studies, the distal cues were located opposite the goal; therefore, participants may not have considered them relevant and/or may have ignored them completely.

To rule out this possibility, we modified Sansa et al.’s [28] task by making two changes. First, we reduced the number of landmarks from four to two. Second, we placed these two cues on the same side as the hidden target, rather than having cues on the opposite side of the arena. However, the proximity of these cues to the target was different, with one cue placed closer to the target compared to the second. By having the two cues relatively close to the target and minimising the number of cues that are available, we hoped that all cues would be readily available to the participant to form an association. We hypothesised that we would see an overshadowing effect, but we asked whether reducing the cue distance would show an overshadowing effect similar to Sansa et al. (i.e., the near cue still gaining more control over performance compared to the far one) or would the overshadowing effect be more equal between the two cues (i.e., no effect of proximity).

## 2. Experiment 1: Testing the Salience of the Cues

In the first experiment, we wanted to make sure that the two cues used in our overshadowing paradigm were equally salient and that participants could use both to find the target. As such, although the cues were different in appearance, we located both cues at the same distance from the target.

### 2.1. Methods

#### 2.1.1. Participants

Participants (n = 12) with a mean age of 24.8 (range 19–46) were recruited using a convenience method of sampling. There were seven females and five males. All participants gave informed consent prior to starting the experiment and were fully debriefed afterwards. Every participant had normal or corrected-to-normal vision. All participants reported being healthy and not having any medical or psychological issues.

#### 2.1.2. Spatial Navigation Task

The spatial navigation task used in the experiment was the NavWell task [34], a human equivalent of the Morris water maze task [35]. The task required participants to virtually navigate a circular arena in order to locate an invisible target. Once the target was found, participants were required to recall this location and navigate to the target on each of the subsequent trials as quickly as possible. The virtual maze was a circular environment (taking 15.75 s to traverse the arena, calculated at 75 Vm along the diameter) with an invisible target (15% of the total arena size) located in the middle of the northeast quadrant. The target remained in the same location for all trials (see Figure 1A inset, black broken square) and only became visible when the participant crossed it. Depending on the experimental condition, one cue (either a small circular light or a green square) was placed on the north wall of the arena (at approximately 25 Vm from the target—Figure 1A inset, yellow circle and green square, respectively).

#### 2.1.3. Procedure

To examine whether the two cues were equally salient, participants were divided into two groups and tested with either the circular light (n = 6) or the green square (n = 6). Experiments were generally conducted between the hours of 9 am and 1 pm. All participants underwent 12 training trials on the virtual water maze task. Participants started each trial from one of four positions around the arena (N, S, E, and W) in a pseudorandom fashion. All participants were required to locate the target within 60 s; if they didn’t locate the target within this time, they were teleported to the goal location. Participants remained at this location for 10 s and were instructed to look around the environment. The time taken to locate the target (in seconds) was recorded for each participant for each trial to measure learning.

Following the learning phase, participants were asked to sit quietly for 5 min before recall was tested. To test recall, a single probe trial was given. Participants were given a single 60 s trial to navigate towards the goal. However, the goal was no longer present. All participants started from a novel position (SW). The percentage of time spent (of the 60 s) in the goal quadrant (NE) compared to the other three quadrants was used to measure recall.

#### 2.1.4. Ethical Considerations & Data Analysis

All experiments were approved by the Maynooth University Ethics Committee (BSRESC-2021-2453422) and conducted according to the ethical guidelines provided by the Psychological Society of Ireland (PSI). The means and standard error of means (SEMS) were calculated for each trial and for each group. Mixed factorial ANOVAs were used to analyse the learning and recall phases. Where relevant, the Tukey HSD test was used for between-group *post-hoc* comparisons, and Bonferroni-corrected *t*-tests were used for within-group comparisons. A star-based level of significance was used where * *p* < 0.05, ** *p* < 0.01, and *** *p* < 0.001.

### 2.2. Results

For the learning phase, a 2 × 12 mixed factorial ANOVA was conducted to compare the time taken to reach the target for both groups across the 12 trials. An overall main effect for the trial was found (F(+11, 99) = 5.778, *p* < 0.001, η^2^ = 0.391). Bonferroni-corrected *t*-tests showed that both groups learned the task, with a significant decrease in time taken to reach the target from trial 1 (mean = 38.7 s, SD = 22.9) to trial 12 (mean = 9.8 s, SD = 3.4). There was no main group effect (F(1, 9) = 1.2, *p* = 0.302) and no significant trial × group interaction effect (F(11, 99) = 1.192, *p* = 0.332) was noted (Figure 1A).

For the recall phase, a 2 × 4 mixed factorial ANOVA was conducted to compare the mean percentage time (of the 60 s) spent by both groups in each of the four quadrants. An overall main effect for quadrant was found (F(3, 30) = 26.56, *p* < 0.001, η^2^ = 0.727). Bonferroni-corrected *t*-tests showed that both groups spent significantly more time in the target NE quadrant (mean = 72.3%, SD = 22, *p* < 0.001) compared to the other three quadrants (Figure 1B). No main group (F(1, 10) = 1.0, *p* = 0.341) or quadrant × group interaction effect (F(3, 30) = 0.384, *p* = 0.765) was noted.

**Figure 1 brainsci-13-01287-f001:**
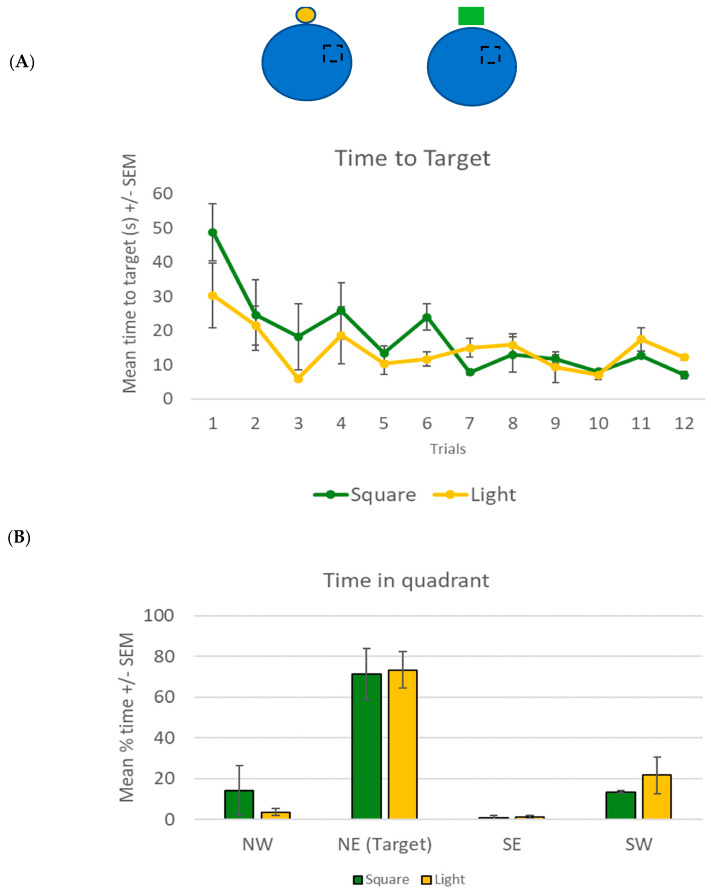
(**A**) Mean time (+/− SEM) taken by participants using the light (yellow circle) or the square (green) to find the hidden target. The hidden target was located on the floor in the NE section of the arena (broken square), with the cue located on the northern wall (see inset for a representative diagram). (**B**) Mean percentage time (of 60 s) +/− SEM spent in all quadrants of the arena (including the target NE) for both groups in the recall trial in experiment 1.

## 3. Experiment 2: Testing the Overshadowing Effect

Results from experiment 1 suggested that both cues were equally salient and that the two cues acquired the same control over the participants’ performance. As such, we were able to move onto the main overshadowing experiment using the same two cues but manipulating the distance of each cue relative to the target.

### 3.1. Methods

#### 3.1.1. Participants

Participants (n = 113) aged 18–46 (mean 21.6, SD = 4.4) were again recruited using convenience and snowball sampling and consisted mainly of Maynooth University students. These included 26 males, 81 females, and 5 participants who did not report. All participants gave informed consent prior to starting the experiment and were fully debriefed afterwards. Some participants from Maynooth University received course credit for their participation. Every participant had normal or corrected-to-normal vision. Participants who self-reported having severe visual impairments, a history of psychological/neurological impairment, a history of motion or simulator sickness, epilepsy, memory issues, a history of drug or alcohol abuse, or were taking psychoactive medication were excluded from the study. *A priori* power calculations were done to estimate the number of participants required to determine the main effect of overshadowing. Using fixed effects ANOVAs and an effect size of 0.3 (see [36]) with a power of 0.9, *p* = 0.05, and five groups (see below), we estimated 114 participants. Furthermore, to ensure participants were generally matched on visual attention and visual–spatial and executive functioning, the Trail Making Test parts A and B were administered [37,38]. In part A, the participant was required to connect numbered circles in ascending order as quickly as possible, and in part B, a letter was introduced (1-A, 2-B, 3-C, etc.). Participants again had to connect the circles containing these numbers and letters in ascending order while they were timed by the experimenter. Lower time scores tend to reflect better performance [39].

#### 3.1.2. Spatial Navigation Task

The Navwell task was again employed in this experiment. The setup was exactly as described above. The target remained in the same location for all trials (again, in the middle of the NE quadrant) and only became visible when the participant crossed it (Figure 2A). However, depending on the experimental group (see Procedure below), one or two cues were used. The cues were located on the wall of the arena and consisted of a small circular light (on the northwest quadrant wall at a distance of 40 Vm (far cue) from the target) and a green square (on the northeast quadrant wall at a distance of 12 Vm from the target (near cue), see Figure 2A,B). The distance between the two cues was approximately 53 Vm.

#### 3.1.3. Procedure

For the learning phase, all participants underwent 12 training trials in the virtual water maze task (see above for details). Again, experiments were generally conducted between the hours of 9 am and 1 pm. Time taken to locate the target (in seconds) and path length (distance travelled in virtual metres [Vm]) were recorded for each participant for each trial to measure learning.

Following the learning phase, participants were asked to complete the Trail Making Test (TMT) to ensure a time delay between the learning and recall phases, as well as to examine whether our groups were generally matched cognitively. The time difference between parts A and B of the TMT was used.

To test recall and the overshadowing effect, a single probe trial was given. Participants were given a single 60 s trial to navigate towards the goal. However, the goal was no longer present. All participants started from a novel position (SW). To examine the overshadowing effect, the cues either remained the same as in the learning phase or a cue was removed for the recall trial. Participants were randomly assigned to one of five different groups: Group 1 (Light–Square/Light, n = 21) was trained with two cues (light and green square) and was retested with one cue (light). Group 2 (Light–Square/Square, n = 24) was trained with two cues (light and green square) and was retested with one cue (green square). Group 3 (Light/Light, n = 23) was trained with one cue (light) and was retested with the same cue (light). Group 4 (Square/Square, n = 23) was trained with one cue (green square) and retested with the same single cue (green square). Group 5 (Light–Square/Light–Square, n = 22) was trained with two cues (light and green square) and was retested with the same two cues (light and green square). See Figure 3A for a visual representation of the groups and locations of the cues. The percentage of time spent in the goal quadrant (NE) was used to measure recall.

Mixed factorial ANOVAs were again used to analyse learning. Where relevant, the Tukey HSD test was used for between-group *post-hoc* comparisons, and Bonferroni-corrected *t*-tests were used for within-group comparisons. A star-based level of significance was used where * *p* < 0.05, ** *p* < 0.01, and *** *p* < 0.001. One-way ANOVAs were conducted to analyse recall as well as to compare age and TMT scores across the five groups.

### 3.2. Results

An initial one-way ANOVA was conducted to determine if age differed across the five groups. Although an overall significant difference was noted (F(4, 104) = 2.920, *p* = 0.025, η^2^ = 0.19), *post-hoc* comparison tests using the Tukey HSD test failed to indicate where those differences lay. A further one-way ANOVA was used to examine whether the five groups were generally well matched on the TMT task; no overall significant difference was noted between the groups (F(4, 103) = 1.633, *p* = 0.172), suggesting that the groups were generally matched for cognitive abilities (see Table 1 for details).

#### 3.2.1. Learning Phase

A 5 × 12 mixed factorial ANOVA was conducted to analyse the time taken to reach the target during the learning phase (Figure 2B). An overall main effect for the trial was found (F(11, 98) = 33.27, *p* < 0.001, η^2^ = 0.78). Bonferroni-corrected *t*-tests showed that all groups had learned the task with a significant reduction in time taken to reach the target from trial 1 (mean = 41.1 s, SD = 19.9) to trial 12 (mean = 12.26 s, SD = 14.3). There was also a significant group effect (F(4, 108) = 7.695, *p* < 0.001, η^2^ = 0.22), with Tukey *post-hoc* revealing that group Light/Light was significantly slower at reaching the target location compared to the other groups (*p* < 0.01, Figure 2B). No significant trial X group interaction effect (F(44, 404) = 1.436, *p* = 0.040) was noted. No significant difference was noted between any group on trial 12 (F(4, 108) = 0.468, *p* = 0.759).

**Figure 2 brainsci-13-01287-f002:**
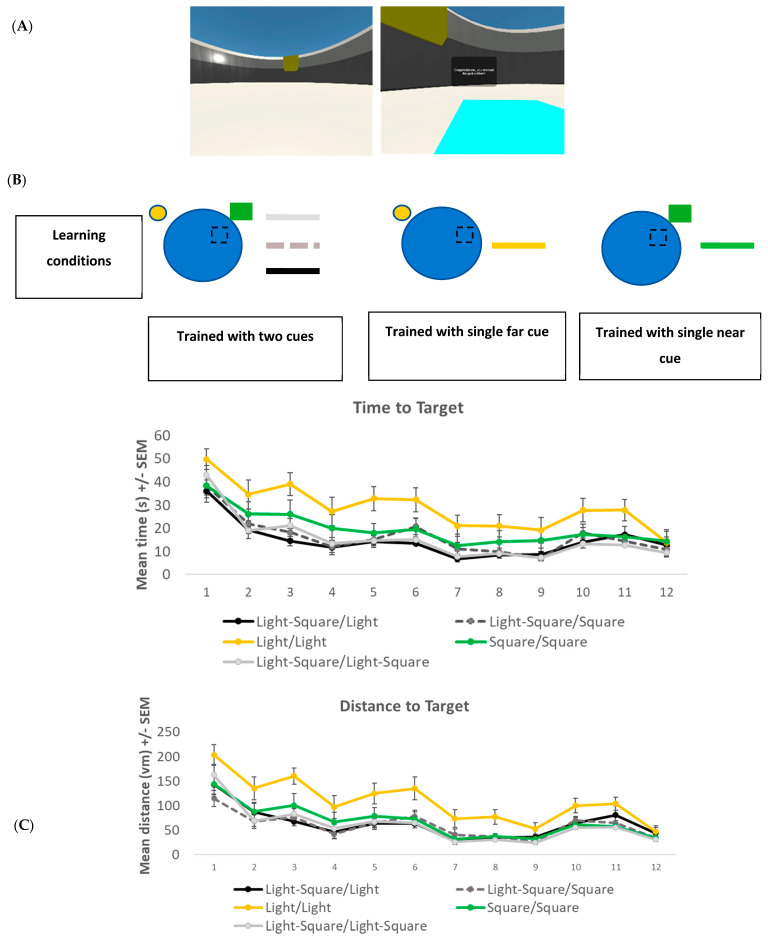
(**A**) Screenshot of the NavWell environment and location of cues—circular light (far) and green square (near)—used in experiment 2. The goal platform (light blue square) becomes visible when it is crossed. (**B**) Schematic representation of learning conditions and mean time in seconds (s) +/− SEM it took to reach the target for five groups across 12 learning trials in experiment 2. (**C**) Mean distance to target (path length) in virtual metres (Vm) +/− SEM for each of the five groups across 12 learning trials in experiment 2.

A further 5 × 12 fixed factorial ANOVA was conducted to analyse the distance taken to reach the target during the learning phase (Figure 2C). Again, an overall significant effect for trial (F(11, 98) = 58.48, *p* < 0.001, η^2^ = 0.87) was found with Bonferroni-corrected *t*-tests, revealing that all groups took significantly shorter paths to reach the target on trial 12 (mean = 37.9 Vm, SD = 34.7) compared to trial 1 (mean = 153.2 Vm, SD = 91.5). Again, a significant group effect (F(4, 108) = 11.91, *p* < 0.001, η^2^ = 0.306) was found, with Tukey *post-hoc* revealing that group Light/Light took significantly longer paths compared to the other groups (*p* < 0.01, Figure 2C). No significant trial X group interaction effect (F(44, 404) = 1.401, *p* = 0.052) was noted. To check that all groups were equivalent on trial 12, a one-way ANOVA was conducted. No significant difference was noted (F(4, 108) = 0.903, *p* = 0.465).

#### 3.2.2. Recall Phase

An overall one-way ANOVA was conducted to compare the mean % time spent by each group in the target quadrant (Figure 3B). An overall significant effect (F(4, 107) = 10.504, *p* < 0.001, η^2^ = 0.39) was found. Tukey *post-hoc* tests showed that the Light–Square/Light group spent significantly less time in the target quadrant compared to all other groups. When we compared each group to chance levels (25%), all groups were significantly different (all *p* < 0.001) except for the Light–Square/Light group (*t*(20) = 0.485, *p* = 0.316). This suggested that, although trained with two cues, this group was unable to use the single cue provided (light) to locate the target.

**Figure 3 brainsci-13-01287-f003:**
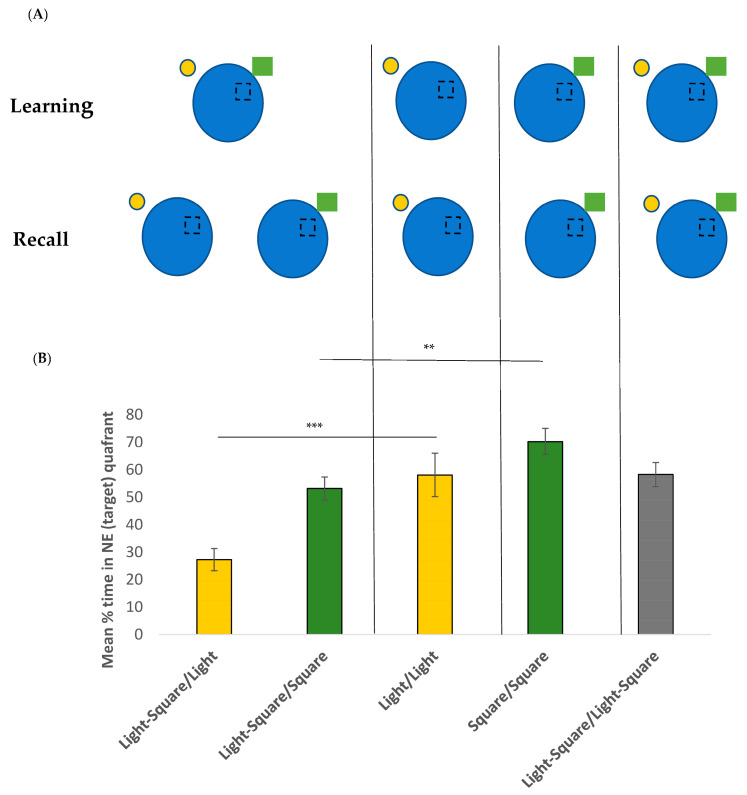
(**A**) Schematic representation of learning and recall groups used in experiment 2. (**B**) Mean % of time (of 60 s) +/− SEM spent in the target NE quadrant for the recall trial for each of the five groups. ** *p* < 0.01 and *** *p* < 0.001.

To specifically examine the overshadowing effect, we first compared the Light–Square/Light group and the Light/Light group (Figure 3B, yellow bars). A significant main effect was found (F(1, 42) = 14.08, *p* < 0.001, η^2^ = 0.25), suggesting that those trained with the two cues but retested with the light only (Light–Square/Light group) spent significantly less time in the target quadrant (mean = 27.3%, SD = 21.6) compared to the Light/Light group (i.e., those trained and retested with the same light cue, mean = 58.1%, SD = 31.4). This suggests an overshadowing effect. We next compared the Light–Square/Square group and the Square/Square group (Figure 3B, green bars). Again, a significant main effect was found (F(1, 45) = 9.67, *p* = 0.003, η^2^ = 0.17) suggesting that those trained with the two cues but retested with the square only (Light–Square/Square group) again spent significantly less time in the target quadrant (mean = 53.2%, SD = 19.8) compared to the Square/Square group (i.e., those trained and retested with the same square cue, mean = 70.3%, SD = 26.3). This again suggests an overshadowing effect.

Finally, an overall significant main effect for proximity was found, whereby the two groups retested with the near cue (Light–Square/Square group and Square/Square group) spent significantly longer time (mean = 61.5%, SD = 20) searching in the target area compared to the two groups retested with the far cue (Light–Square/Light and Light/Light, mean = 43.4%, SD = 31, F(1, 87) = 15.207, *p* < 0.001, η^2^ = 0.149).

## 4. Discussion

Here we replicate recent findings by Sansa et al. [28] and show that overshadowing occurs during spatial learning in humans and that landmarks located closer to the goal control searching behaviour more than those located further away. This was contrary to our prediction, where we thought that the location of the cues would allow the overshadowing effect to be more equal and that no effect of proximity would be found. This suggests that absolute distance is not a factor. While there are multiple studies demonstrating general overshadowing effects across a range of species, including honeybees [11], birds [12], and other nonhuman mammals [10,13], up until recently, there have been only a few studies examining this in humans. These results add to the growing literature suggesting associative learning provides a good account for spatial memory in the water maze task. With associative learning theories, there is recognition that only certain elements of the environment are used during learning and that not all elements are treated equally. We demonstrated this by showing that the near cue (the most salient one) is used, and this cue overshadows the far one through competition.

Sansa et al. explained their overshadowing effects in terms of generalisation decrement [40,41] rather than associative competition because the authors argued that participants could approach the goal from different directions across the trials; therefore, all of the equally-spaced cues could be learned (see also [42]). However, given that performance was better controlled by the proximal cues compared to the distal ones, cue location may have been a factor in their results. Similarly, Commins et al. [34] using a virtual water maze task also showed that participants were slower to learn the target and subsequently were not as accurate in recalling the location when cues were placed opposite the target compared to those that were placed near the target. We tried to rectify this and had the cues more prominently placed (i.e., both were on the same side as the target); despite this, the near cue still controlled behaviour significantly more than the far cue. Although we cannot rule out the possibility of generalisation decrement as an explanation—a one-cue environment (as during retest) is different from a two-cue one (as during training)—we offer an alternative interpretation. Examination of both our findings and those of Sansa et al. [28] shows that some cues are totally ignored. For example, when retested with just the two distal cues (Sansa et al.) or the far cue (light, current experiment) after being trained with the compound set, participants searched in the target location at the chance level (25%) only. This would suggest that during training, when both far and near cues were available, participants learned the location relative to the near cue(s) and completely ignored the far and less salient one(s). Interestingly, Herrera et al. [33] ruled out the possibility that participants ignored the distal landmark in their overshadowing experiment, pointing out that their discontiguous group (geometry + distal cue) learned the task better than the control group (geometry only). However, in both the current and Sansa et al.’s [28] studies, participants in the proximal + distal groups learned the task equally well as those in the proximal group alone. Therefore, ignoring the less salient distal cue during training is a strong possibility here, which led to searching at chance levels during retest.

Although all cues can provide information on the target location (experiment 1), participants seemed to learn about the most salient one only, i.e., the nearest one in the current experiment. This perhaps speaks to the idea of ‘learning efficiency’ (see [43]), whereby participants use the least effortful way of learning. For example, if two cues are in a compound, they don’t learn the location of the target with respect to both cues (as this may be effortful) or indeed the less salient one (as again, this may be effortful), but simply learn with respect to the more salient one. There is good reason to believe that learning the location relative to the distal cue is more effortful because we showed that participants were significantly slower to learn the task when only the far cue was available (Group Light/Light, Figure 2B,C; see also [34]). Similarly, in Sansa et al.’s study, the group trained with the distal cues only (Group 2-2D, Figure 3) was also significantly slower to learn the target location compared to the other groups. This suggests that although the location of the platform can be learned relative to the distal cue(s), it is more effortful. As such, it is ignored, especially when presented alongside a more salient cue. Future experiments should examine whether increasing the salient value of the more distal cues (e.g., having them bigger or brighter) might allow for this cue to be used during learning. In addition, as the virtual water maze task is a very simple one, these findings should be tested further in a more complex and ecologically valid environment.

In conclusion, reducing the number of cues and having them both closer to the target location (to ensure that all cues are noticed) resulted in an overshadowing effect whereby the near cue controlled performance more than the far cue. However, rather than the far cue being simply overshadowed, it seemed to be totally ignored. We suggest that participants use a learning strategy that tries to reduce the amount of effort needed by learning locations relative to only the most salient of cues.

## Figures and Tables

**Table 1 brainsci-13-01287-t001:** Demographic information for each experimental group in the overshadowing experiment (experiment 2).

Group	N	Age	M/F/Not Reported	TMTb-a
Light–Square/Light	21	22.5 (1.6)	4/15	21.1 (2.6)
Light–Square/Square	24	23.2 (1.01)	6/17	21.7 (2.0)
Light/Light	23	22.5 (1.7)	10/13	24.8 (5.9)
Square/Square	23	19.9 (0.35)	4/19	16.2 (2.4)
Light–Square/Light–Square	22	19.9 (0.3)	3/17/2	16.6 (1.7)

## Data Availability

The data presented in this study are available on request from the corresponding author.

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
