# Peer review of "Landmark Distance Impacts the Overshadowing Effect in Spatial Learning Using a Virtual Water Maze Task with Healthy Adults"

_brainsci, 2023, doi:10.3390/brainsci13091287_

Round 1

Reviewer 1 Report

The authors presented interesting data about overshadowing effects in spatial learning in humans. Please consider the following minor requests:

1. Please indicate the number of participants per sex (line 190).

2.  Please indelicate the time when experiments were performed.

3. Fig. 2. In order to achieve better contrast, please choose some other colors for Light-square/Light-square and Light-square/Square groups.

 4. Fig 3. Please indicate in legend the meaning of asterisk.

Reviewer 2 Report

Review of MS # brainsci-2566241: Landmark distance impacts the overshadowing effect in spatial learning using virtual water maze task with healthy adults, by Deery & Commins

The conclusion ("We suggest that participants use a learning strategy that tries to reduce the amount of effort needed by learning locations relative to only the most salient of cues") is an overinterpretation which is not relevant for the process of cognitive mapping.  Specifically, the study is based on reaching a target in a virtual environment by means of two landmarks. This is a spatial navigating task for which a representation of the environments is not necessary. It is more a sort of orienting in reference to a beacon, where another information is unnecessary. This is not what O'keefe and Nadel considered as cognitive mapping (1978; p. 242): “the hippocampal locale system is assumed to form the substrate for maps of environments an animal has experienced; these maps are established in the hippocampus during exploration, a species-specific behaviour pattern concerned with the gathering of information”. Accordingly, the argument against cognitive mapping seems irrelevant for this task and ought to be diminished. In other words, stating that "These results add to the growing literature suggesting associative learning provides a good account for spatial memory rather than the need to evoke a cognitive map" (L. 340-341) is an over-interpretation when based on searching one goal based on 1-2 landmarks.

Another concern is methodological: what was the size of the arena (22Vm diameter?), and what was the distance between the landmarks in Vm, as well as their distance to the target? Things became vague when talking only in proximal and distant cues. Traditionally, proximal cues are within easy-reaching distance whereas distal ones are those in the background (panorama), with the latter considered more stable. Here these terms are used without informing what makes a distal or a proximal landmark. 

The text is repetitive and the description of the experiments in the abstract, introduction, method and results, unnecessarily impose on getting to the most conclusive finding, which is nicely described in L.311-316. Removing repetitions and surplus information will highlight the finding on over-shadowing, which somehow gets lost in detail (e.g. – after the first mentioning, use only 'light' and 'green square' and skip the NE, NW).

Other comments:

1.        Remove references, n's of groups, NavWell, etc. from the abstract.

2.        L.317-323: The difference between Light-Square/Square group and Square/Square groups (Figure 3B, green bars) is not convincing. What is the real difference in seconds (not percentage)? – could be just a 2-3 sec difference? If so, this is not a solid base for this result.

3.        The text of Experiment 2 is puzzling: a part of the results (Figure 2) appears before the procedure? Also: are both figures 2b and 3c required (they are similar, perhaps since subject traveled in the same speed).

4.        In Exp 2 – retesting with the distant light signal could simply awarded less accurate information regarding the location of the correct quadrant. Could it be as simple as that (no need for the overshadowing argument)?

L.357-360: If so, the difference between the present study and Sansa et al. is merely interpretive: some cues are totally ignored rather that generalization decrement as an explanation.

.
